# Emergent and Predictable Memorization in Large Language Models

**Stella Biderman**[1,2], **USVSN Sai Prashanth**[2], **Lintang Sutawika**[2], **Hailey Schoelkopf**[2,3], **Quentin Anthony**[2,4], **Shivanshu Purohit**[5], **and Edward Raff**[1,6]

[1]Booz Allen Hamilton, [2]EleutherAI, [3]Yale University, [4]Ohio State University, [5]Stability AI, [6]University of Maryland, Baltimore County

## Abstract

Memorization, or the tendency of large language models (LLMs) to output entire sequences from their training data verbatim, is a key concern for deploying language models. In particular, it is vital to minimize a model's memorization of sensitive datapoints such as those containing personal identifiable information (PII). The prevalence of such undesirable memorization can pose issues for model trainers, and may even require discarding an otherwise functional model. We therefore seek to predict which sequences will be memorized before a large model's full train-time by extrapolating the memorization behavior of lower-compute trial runs. We measure memorization in the Pythia model suite and plot scaling laws for forecasting memorization, allowing us to provide equi-compute recommendations to maximize the reliability (recall) of such predictions. We additionally provide further novel discoveries on the distribution of memorization scores across models and data. We release all code and data necessary to reproduce the results in this paper at https://github.com/EleutherAI/pythia.

## 1 Introduction

Recent natural language processing (NLP) research in generative tasks has largely been driven by two findings: (1) the transformer architecture performs well [Vaswani et al., 2017, Devlin et al., 2018, Radford et al., 2019]; and (2) increasing the scale of transformer architectures leads to improved performance [Brown et al., 2020, Chowdhery et al., 2022]. In addition to these benefits, transformers are a general and multipurpose architecture that have achieved state-of-the-art results outside of NLP on diverse tasks such as text-to-image synthesis [Ramesh et al., 2022, Crowson et al., 2022, Rombach et al., 2022], code generation [Chen et al., 2021, Xu et al., 2022, Fried et al., 2022], and protein modeling [Jumper et al., 2021, Ahdritz et al., 2022]. Despite their widespread success and increasing use, the learning dynamics of transformer models are poorly understood and research into how a given model learns and internally represents data has the potential to affect a broad range of high-impact applications. As these models become increasingly adopted, it is essential that we develop better and more reliable tools for measuring and controlling their behaviors.

### 1.1 Memorization in Large Language Models

In particular, the demonstrated capacity and ability of these large language models to memorize data has become a significant concern [Carlini et al., 2019, 2021, Hu et al., 2022]. The most obvious ramification is personal information or otherwise sensitive data being leaked to the public at large and extracted by a bad actor. This has motivated extensive research into mitigating memorization

---

Correspondence to: `stellabiderman@gmail.com`

37th Conference on Neural Information Processing Systems (NeurIPS 2023).

by decreasing the *total quantity of memorized text* [Lee et al., 2021, Kandpal et al., 2022, Carlini et al., 2022, Hernandez et al., 2022]. Although this is an admirable goal, we ultimately do not view it as an appropriate solution as **only some memorization is bad**. What's more, **some types of memorization are good**: we want large language models to memorize factual events and details to avoid "hallucinating" plausible-sounding but errant facts to unsuspecting users [Power et al., 2022, Cao et al., 2022, Tirumala et al., 2022b].

Despite the extensive literature on memorization in trained models [Carlini et al., 2019, 2021, Hu et al., 2022], there are few tools to help practitioners either prevent memorization or detect it early in model training. Before the advent of transformer-based large language models, using differential privacy was popular [Abadi et al., 2016, McMahan et al., 2017, Popov et al., 2017]. However, such methods have been observed to hurt performance during pretraining [Anil et al., 2021], and are therefore not popular among people who train large language models. In recent years, the bulk of interventionist work has focused on how removing duplicated samples from the training dataset can decrease memorization [Lee et al., 2021, Kandpal et al., 2022, Carlini et al., 2022, Hernandez et al., 2022]. Importantly, these works focus on memorization *on average* and cannot be relied on to prevent memorization of specific training examples. Additionally, Biderman et al. [2023] shows that even when models are trained on deduplicated data they can still memorize substantial amounts of their training corpus.

Another approach to constraining model behavior is to intervene at inference time instead of at training time. Ippolito et al. [2022] introduce an interference-time intervention that has a 100% success rate at preventing *verbatim* memorization, but they note both that their methodology is easily subverted, and does not fulfill the intention behind the term "memorization," which is the tendency of models to learn entire samples *during training* without understanding their underlying meaning. We view test-time intervention as a promising avenue for future research, especially due to its success in other domains [Haghighatkhah et al., 2021, Ravfogel et al., 2022, Belrose et al., 2023a,b], but it is insufficent as a solution because some of the concern regarding PII and copyrighted text concerns *instrinsic properties of the model* and not merely its output text. Although at first glance these techniques may seem more viable as downstream users can apply them to API models, most models offered via an API do not give sufficient access to the user to apply them. A missing component of the memorization literature is an investigation of *which specific data points are memorized* in large language models.

## 1.2   Scaling Laws

Due to the substantial cost of training large language models, it is highly desirable to be able to make predictions about model characteristics before they are actually trained. The literature on *scaling laws* [Kaplan et al., 2020, Henighan et al., 2020, Hernandez et al., 2021, Mikami et al., 2021, Hoffmann et al., 2022] has been successfully used to inform the decision-making of a variety of researchers at model training-time by allowing them to generalize the decisions made while investigating smaller models to inform the design of larger (sometimes by many orders of magnitude) models [Rae et al., 2021, Black et al., 2022, Scao et al., 2022b, Chowdhery et al., 2022]. While this work on scaling laws does extend to memorization [Carlini et al., 2022, Hernandez et al., 2022], how memorization evolves during a model's training process across a variety of scales has not been studied.

Scaling law studies are typically used to find a model with the best loss trainable within a given computational budget. However, the intuitions and experimental design around cheaply estimating the qualities of a final model can be applied to more than just training loss–we would like practitioners to be able to extrapolate other more complex properties of their final trained model prior to performing the costly training run.

In the case of memorization, because some data points are far more undesirable for a model to memorize, such as PII, it would be desirable for engineers to be able to predict whether a model will successfully avoid memorizing such harmful data, and make informed risk assessments and decisions about training prior to large compute expenses.

In our work, we study the creation of tools to predict the memorization of *specific data points* prior to model training, rather than the macro-level corpus-wide statistics considered by prior work. We take a first step in this direction by proposing two strategies: 1) making predictions from a smaller model to a larger model; and 2) making predictions from a partially trained model of a given size to the fully

trained model. Using smaller or partial model training runs to inform large model training runs is critical, because these runs provide a cheap method to inform training behavior for a given corpus, rather than training an entire large model from scratch. We find that the efficacy of our proposed methods varies with their downstream intended use, depending on whether precision (we want to confirm something was memorized) or recall (we want something "forgotten") is most desired.

### 1.3 Our Contribution

In this paper, we introduce the question of extrapolating a model's memorization behavior for *specific training data points* based on evaluations set in relatively low-cost training regimes. These low-cost regimes enable us to abort models without wasting significant compute resources in the event of undesirable behavior. This includes the *typical setting*, where we extrapolate the qualities of large models based on small models, as well as a *novel setting* where we extrapolate the behavior of fully-trained models based on partially-trained models. As far as we are aware, we are the first paper to study forecasting model behavior in this novel setting.

Our primary contributions are:

1. Introducing the problem of forecasting whether or not a model memorizes a *specific* training datum.

2. The discovery that the memorization of a specific training string by a large language model is not reliably predicted by either studying smaller language models or partially trained checkpoints, unless a sizable fraction of the pretraining compute of the target model is used.

3. A preliminary analysis of scaling laws for forecasting memorization, and recommendations for maximizing forecast reliability given a set compute budget to make this prediction.

### 1.4 Roadmap

The rest of this paper is organized as follows: in Section 2 we present relevant facets of our methodology, including definitions of metrics (Sections 2.1 and 2.3), the threat model (Section 2.2), and choice of pretrained models (Section 2.4). In Section 3, we explore the feasibility of predicting the memorization behavior of large models based on small models. Further, in Section 4, we explore the feasibility of predicting the memorization behavior of the fully-trained model based on intermediate checkpoints. We then analyze these two kinds of predictors head-to-head and plot scaling behavior in Section 5 We perform ablations to confirm our method is robust to thresholding choices in Appendix A and to deduplication in Appendix B.

## 2 Methodology

### 2.1 Measuring Memorization

| Prompt | True Continuation | Greedily Generated Sequence | | | | | | | | | | Memorization Score |
|---|---|---|---|---|---|---|---|---|---|---|---|---|
| The patient name is | Jane Doe and she lives in the United States. | John | Doe | and | he | lives | in | the | United | Kingdom | . | $\frac{0+1+1+0+1+1+1+1+0+1}{10} = 0.7$ |
| Pi is defined as | the ratio of the raidus of a circle to its | a | famous | decimal | that | never | enters | a | repeating | pattern | . | $\frac{0+0+0+0+0+0+0+0+0+0}{10} = 0$ |
| The case defendant is | Billy Bob. They are on trial for tax fraud | Billy | Bob | . | Are | they | really | on | trial | for | tax | $\frac{1+1+1+0+0+0+0+0+0+0}{10} = 0.3$ |
| The case defendant is | Billy Bob. They are on trial for tax fraud | Billy | Bob | . | They | are | on | trial | for | tax | fraud | $\frac{1+1+1+1+1+1+1+1+1+1}{10} = 1$ |

Table 1: Examples of memorization score calculation with different prompts. Note that these are provided for illustrative purposes and are not from the actual training data. The final example demonstrates a 4-extractible string.

"Memorization" is typically cast as a form of overfitting or failure to generalize outside the training data distribution, distinct from "good learning" in some senses. However, formalizing this intuition presents challenges. In this paper, we consider the framework introduced by Carlini et al. [2021] grounded in *k-extractibility*:

**Definition 2.1.** A string $s$ is said to be $k$-extractible if it (a) exists in the training data, and (b) is generated by the language model by prompting with $k$ prior tokens.

To demonstrate, the training data sequence "*Their email address is me@alice.com*" is 3-extractible (memorized) if the prompt "*Their email address*" yields "*is me@alice.com*"—thus producing an

exact copy of the training data sequence. We term the accuracy of tokens in the continuation as the *memorization score* of the sequence and call a sequence ($k$-)memorized or ($k$-)extractable if the memorization score is 1. Illustrative examples are shown in Table 1

$$score(M, N) = \frac{1}{N} \sum_i^N 1(S_{M+i} = G_{M+i}) \tag{1}$$

In addition to $k$-extractability, we evaluate the *memorization score*, defined as the number of ordered matching tokens between the model's greedily generated sequence $G_{32:64}$ and the dataset's true continuation $S_{32:64}$ of a sequence $S \in D$ on a given prompt. See Equation (1) for the formal equation, where $N$ is the length of the true continuation and greedily generated sequence (32 in our case), and $M$ is the length of the prompt (also 32 in our case). A *memorized* or *extractable* sequence has a memorization score of 1.

Doing a forward pass on a large transformer is relatively expensive, costing about one third the cost of a full gradient update step. Consequently, feeding the full training data through the model for a forward pass would cost approximately one third the amount of compute that training the model did, and doing the full seven checkpoints that we do would come out to a larger compute budget than training the models themselves.

To ensure computational feasibility in our experiments, we choose $k = 32$ and evaluate the first 64 tokens from each sequence (we verify the robustness of this choice in Appendix A). Each sequence is a set of 2049 tokens, sampled from shuffled documents. These sequences are the input data points to the model during training.

## 2.2 Threat Model

Throughout this paper, we assume that an engineer is looking to train a large language model with billions of parameters on a dataset, and that there is a small subset of the dataset that would be undesirable to have the model memorize. The engineer wishes to accurately predict whether this subset of the training data will be memorized by the fully-trained model, and would like to do so cheaply by expending a relatively small amount of compute on testing prior to the full training run. Following the literature on scaling laws [Kaplan et al., 2020, Hoffmann et al., 2022], the cost in FLOPs of training a model is approximately

$$C = 6 \times \text{[\# Params]} \times \text{[\# Tokens]} \tag{2}$$

and we assume the engineer has a computing budget that allows them to perform substantial testing before performing the full model training run. Because the engineer is the model trainer, We assume that the engineer has full access to the final and test models' training runs, including viewing the ordering and content of data points seen by their model and saving intermediate checkpoints as desired.

We focus on the *prediction* of memorization, rather than simply filtering all data points that contain PII or are otherwise undesirable to memorize, for several reasons. First, although PII can be filtered from LLM training datasets via heuristic filters, or more advanced methods like classifier-based approaches, these filtering methods are not perfect, and may fail to remove certain instances of PII from the dataset [Li et al., 2023].

Additionally, there are also non-PII data points that are undesirable to memorize but desirable to train on, including copyrighted material. Some works suggest such data is essential to achieving high quality performance in certain domains [Longpre et al., 2023, Min et al., 2023]. While the ethical and legal intricacies of permissible training data are outside the scope of this paper, the memorization of such content is a widely contested issue.[1]

## 2.3 Predicting Memorization

We can treat a smaller model's memorization of a sequence, or lack thereof, as a predictor for the memorization behavior of a larger model. Whether the interested model did memorize the sequence is the ground truth label, and the smaller model's behavior is the prediction.

---

[1]For instance, Github Copilot's memorization of training data.

For example, if a smaller model memorized a sequence and the larger model did not, we can think of this case as a false positive. Likewise, if both models memorized the sequence, then the smaller model's prediction was a true positive. Models not memorizing the target sequence are negative cases.

This "prediction" by the smaller model compared against the ground truth allows us to calculate classification metrics such as precision and recall. In this case, *precision* tells us how many of the sequences memorized by the smaller model are also memorized by the larger model. *Recall* conveys the percentage of sequences memorized by the larger model that are also memorized by the smaller model. The same framing can also be applied when analyzing across time—where we compare the memorized sequences at a certain intermediate checkpoint, and wish to predict which sequences will be memorized by the completed model.

As the engineer's sole concern is to avoid memorization on an undesirable subset (see Section 2.2), false negatives and false positives in predicting memorization have very different impacts on their workflow: a false positive (i.e. incorrectly predicting that a model will memorize the undesirable subset) results in throwing away a cheap model that could have been fruitfully continued to train the final model, while a false negative (i.e. incorrectly predicting that a model will not memorize the undesirable subset) results in the costly training of a full model that could leak sensitive samples from the training dataset. We are therefore primarily interested in assessing the *recall* of the predictors and will tolerate a low precision if it comes with a high recall. We explore the tradeoffs in these costs in Section 3.

### 2.4  Choice of Models and Datasets

At the time of writing, the only publicly-available pretrained LLM scaling suites trained on fully public training data are EleutherAI's GPT-Neo [Black et al., 2021, Wang and Komatsuzaki, 2021, Black et al., 2022] and Pythia models [Biderman et al., 2023], and Cerebras systems' Cerebras-GPT [Dey et al., 2023]. All of these suites were trained on the Pile [Gao et al., 2020, Biderman et al., 2022]. Additionally, we were able to obtain access to the ROOTS dataset [McMillan-Major et al., 2022, Laurençon et al., 2022] that the BigScience Workshop's BLOOM [Scao et al., 2022a] model was trained on. Of these model suites, we choose to use Pythia because **(a):** All Pythia models saw data samples in the exact same order, and that order is publicly available, **(b):** the training data differs slightly across the GPT-Neo models, **(c):** some BLOOM models only have three partially-trained checkpoints, and **(d):** Cerebras-GPT models don't provide partially-trained checkpoints. The OpenLlama, RedPajama, and StarCoder [Geng and Liu, 2023, Computer, 2023, Li et al., 2023] models have additionally been trained on public data, but were released after the completion of this work and do not release intermediate checkpoints.

We note that this limitation of eligible open models is not a drawback according to our threat model– we intend to create a novel procedure and tool for risk assessments usable by *model creators*, who have full control and access to information about their model training.

The computational cost of many of the experiments we run is quite large. Consequently, we are unable to evaluate every partially-trained model checkpoint in the Pythia suite.[2] For most of our experiments, we choose to evaluate seven checkpoints spaced evenly throughout training. Specifically, we evaluate on checkpoints trained for $(23 \cdot 10^6)$, $(44 \cdot 10^6)$, $(65 \cdot 10^6)$, $(85 \cdot 10^6)$, $(105 \cdot 10^6)$, $(126 \cdot 10^6)$, and $(146 \cdot 10^6)$ sequences respectively, where these checkpoints approximately correspond to 7 checkpoints evenly spaced throughout training. We use the GPT-NeoX library [Andonian et al., 2021] that trained Pythia to efficiently implement our evaluation protocol.

## 3  Memorization Across Scales

By far, the most common type of scaling law to study (and indeed, the origin of the term itself) is looking at how performance for very large models can be predicted based on performance of much smaller models. Fully-trained smaller model variants are independently useful as artifacts and can be applied in resource-constrained environments in place of larger models. Therefore, when projecting the characteristics of higher-compute model runs via scaling studies, training smaller model variants for this purpose is an actively desirable by-product, in contrast to the alternative of producing many

---

[2]The cost of doing so would be comparable to the cost of training the models in the first place.

shorter-training-duration checkpoints of the same single large architecture to extrapolate properties of a final full run. Therefore, the first question we seek to answer is: can an LLM's memorization behavior be predicted across model scales?

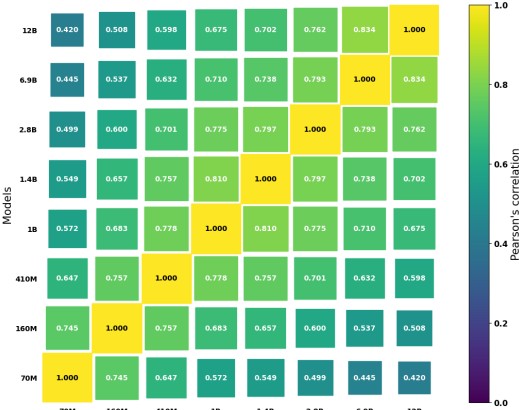

(a) A Hinton diagram for visualizing the correlation between fully-memorized sequences by different model sizes. All models are fully trained.

| Model | Precision | Recall |
|-------|-----------|--------|
| Pythia-70M | 0.956 | 0.197 |
| Pythia-160M | 0.948 | 0.289 |
| Pythia-410M | 0.940 | 0.401 |
| Pythia-1.0B | 0.931 | 0.512 |
| Pythia-1.4B | 0.926 | 0.554 |
| Pythia-2.8B | 0.909 | 0.658 |
| Pythia-6.9B | 0.884 | 0.795 |
| Pythia-12B | — | — |

(b) Precision and Recall when using each model to predict which sequences would be memorized by the 12B parameter model. For example, 95.6% of the sequences memorized by the 70M model were also memorized by the 12B model, but those only accounted for 19.7% of the sequences that the 12B model memorized.

Figure 1: A look at memorization across scales

To evaluate how productive training small models can be for the purpose of predicting which datapoints will be memorized by large models, we subset our data to the sequences with a memorization score of 1 (meaning all 32 target tokens were produced accurately by the smaller model). Then, we look at the correlations between each pair of fully-trained model sizes for which sequences are memorized. The results are shown in Figure 1a.

We see a sharp decline in correlation between which sequences are memorized by smaller models and the 12B model as the gap between the model sizes increases. Unfortunately, we find that these low correlation scores cause the set of sequences memorized by small models to have very poor predictive power in terms of what sequences will be memorized by a larger model. We also measure precision and recall of fully-memorized sequences using each smaller model to predict the memorization of the 12B model as shown in Figure 1b. Although the *precision* is high for all models (see Section 2.2), we are more interested in achieving a high recall than a high precision. The recall is incredibly low across the board, with even the 1.4B parameter model only achieving a recall of $0.554$ when trying to forecast the behavior of a model an order of magnitude larger.[3]

Our findings suggest that using smaller model runs to forecast the memorization of larger models is not accurate. Due to the low recall, practitioners cannot use a small model's lack of memorization of a given sequence as a strong guarantee that their larger model will not memorize that same sequence. We therefore do not recommend using smaller model runs for this task, and seek to provide a setup that grants practitioners more assurances and a better compute tradeoff.

## 4 Memorization Within Training

The second question we seek to answer is: can an LLM's memorization behavior be predicted ahead of time within a training run? We wish to determine if, by testing memorization behavior after partially completing a training run, an engineer can achieve a reliable signal about whether undesirable portions of the training data are memorized and if so to abort a training run early.

Our analysis in this section is motivated by the finding of Biderman et al. [2023] that location within the training data does not impact whether a particular sequence is memorized. Therefore, we hypothesize that those concerned about the memorization of particular strings could move them early

---

[3]Typical use-cases are to use smaller models to predict the behavior of models one to two orders of magnitude larger, see Rae et al. [2021], Scao et al. [2022b], Chowdhery et al. [2022].

during training. Thus practitioners would have an early warning signal for detecting memorization of undesired sequences. Unfortunately, we continue to find largely negative results, but hope that future research with better techniques for predicting memorization might vindicate this idea.

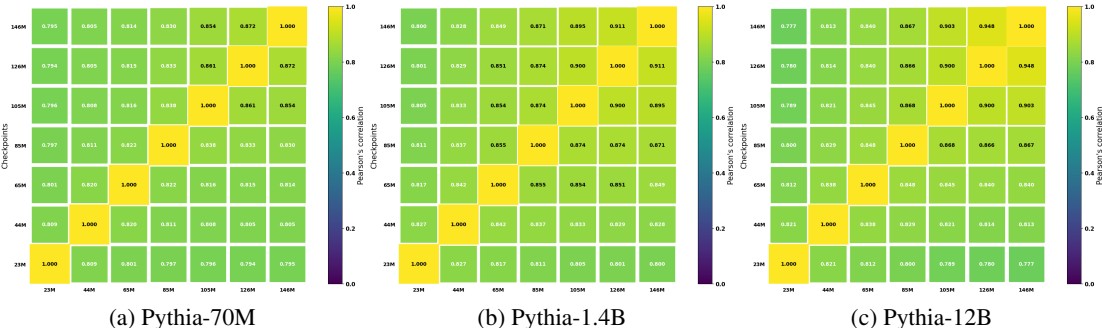

| (a) Pythia-70M | (b) Pythia-1.4B | (c) Pythia-12B |

Figure 2: Heat maps visualizing the correlation between which sequences are memorized by different checkpoints. Plots for other Pythia models can be found in Figure 10.

In Figure 2, we show a correlation heatmap between which sequences are memorized by different checkpoints of the same model. We only look at memorization of the first 23 million sequences, as that is the data that our least-trained model checkpoint has seen.

| Seq Num | Precision | Recall | | Seq Num | Precision | Recall |
|---|---|---|---|---|---|---|
| $23 \cdot 10^6$ | 0.919 | 0.513 | | $23 \cdot 10^6$ | 0.918 | 0.500 |
| $44 \cdot 10^6$ | 0.913 | 0.587 | | $44 \cdot 10^6$ | 0.915 | 0.575 |
| $65 \cdot 10^6$ | 0.910 | 0.658 | | $65 \cdot 10^6$ | 0.913 | 0.641 |
| $85 \cdot 10^6$ | 0.910 | 0.721 | | $85 \cdot 10^6$ | 0.911 | 0.711 |
| $105 \cdot 10^6$ | 0.915 | 0.816 | | $105 \cdot 10^6$ | 0.916 | 0.809 |
| $126 \cdot 10^6$ | 0.945 | 0.918 | | $126 \cdot 10^6$ | 0.943 | 0.916 |
| $146 \cdot 10^6$ | — | — | | $146 \cdot 10^6$ | — | — |
| (a) Pythia-6.9B | | | | (b) Pythia-12B | | |

Table 2: Precision and recall for predicting which sequences would be memorized by the fully-trained model from a partially-trained checkpoint. We observe consistently high precision, but only achieve high recall after significant compute has been expended (later intermediate checkpoints).

Our results on precision and recall (Table 2) largely mirror those of Section 3 in general trends. We see that the earliest intermediate checkpoints we test do not exhibit the high recall that is desirable, for instance with the 23M checkpoint of Pythia-12B underperforming the fully-trained Pythia-6.9B in recall.

We thus observe that using intermediate checkpoints of a model run to predict memorization is not a silver bullet—it is still the case that precision remains high throughout models, but recall is low for all predictors that use significantly less compute than the final model's cost. Therefore, in this setting as well, it is easier to guarantee a sequence *will* be memorized through such extrapolations rather than not. Since the latter guarantee of non-memorization is more useful to engineers, our focus thus shifts to determining the compute-optimal model to train to gain a desired level of recall, in order to maximize predictive power amongst the options we explore.

## 5   Scaling Laws

Having established the empirical results in the previous section, we now examine our results through the lens of computational efficiency and scaling laws, where the aim is to achieve the most reliable results for the least expense. To achieve this, we examine how well models of various sizes and number of training steps predict which sequences will be memorized **by the fully trained 12B**

**parameter model**. This is in notable contrast to Section 4, where partially-trained models are only compared to fully-trained models of the same size. As a visual aid, models with the same size are colored the same.

## 5.1 Unusual Scaling

In the overwhelming majority of prior work on scaling laws [Brown et al., 2020, Kaplan et al., 2020, Pu et al., 2021, Mikami et al., 2021, Rae et al., 2021, Black et al., 2022, Scao et al., 2022b, Chowdhery et al., 2022], including scaling studies targeting memorization [Carlini et al., 2022, Hernandez et al., 2022, Tirumala et al., 2022a], plots of quantities of interest vs. compute are linear on a log or log-log plot. We find that this is not the case in our setup for both precision and recall.

The scaling data for precision is extremely anomalous. Not only are the plots non-linear, we find that the behavior of the 12B partially trained model is extremely out-of-line with the behavior of smaller models. The results for recall are less anomalous, lacking the divergent behavior for the 12B model, but nevertheless do not accord with what the scaling laws literature generally expects.

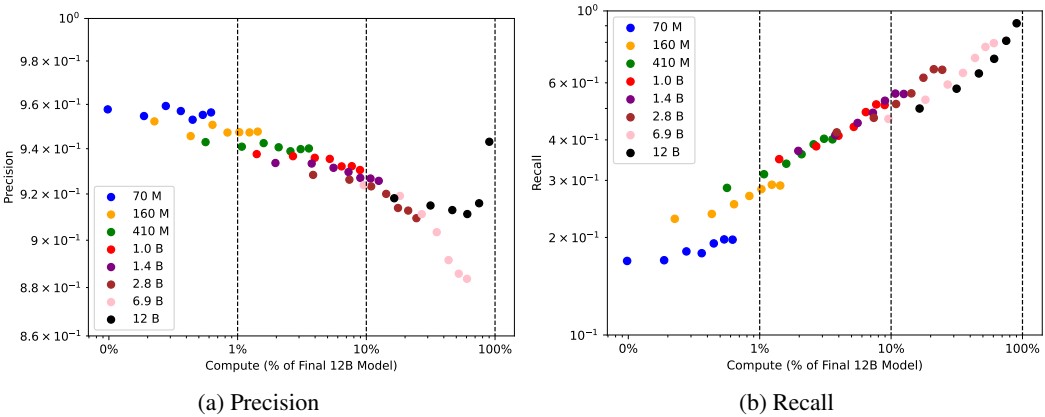

(a) Precision             (b) Recall

Figure 3: Scaling curves for Pythia models.

Despite the fact that there is a high-level pattern in the scaling laws curve for recall, a careful look at the data indicates unusual behavior. In the low-compute regimes, which are of most interest to engineers looking to minimize the cost of creating a prediction of the behavior of large models before they are trained, we see a consistent pattern of larger models being better than smaller models for a fixed compute budget. However, as the amount of compute expended scales, this is no longer the case. Starting at around 1% the budget of the fully trained model, equicompute models perform the same regardless of the number of parameters. Starting at around 10% the budget of the fully trained model, the *smallest* model trained for this compute budget becomes the best predictor of memorization in the fully trained model.

## 5.2 Emergent Memorization

We also see evidence of "emergent" or "semi-emergent" behavior as model scale increases. In the literature on emergent behavior in large language models [Srivastava et al., 2022, Ganguli et al., 2022, Wei et al., 2022, Caballero et al., 2022], the term refers to when a large model's performance on a task is substantially different from the extrapolated value of curves fit to the behavior of smaller models. Often, but not always, this occurs when performance goes from near-zero to meaningful. While our situation is not totally analogous, one can similarly consider "emergent memorization" to occur when data is memorized by large models which cannot be predicted based on the memorization behavior of smaller models. Since, by definition, emergent behavior implies that smaller-scale model behaviors are qualitatively different to those of larger models, this can pose challenges for traditional scaling laws or for extrapolating model behavior to models orders of magnitude larger. As a result, we suggest that this is an important area for further study, including expanding the scope of our work to models larger than 12B parameters.

### 5.3 Takeaways for Engineers

As discussed in Section 2.2, the primary point of interest to engineers is to predict the behavior of a large language model before it is trained. Such predictions should be grounded in low-cost regimes such as the behavior of trained "test" models that are at least an order of magnitude smaller than the target model. We find that for cases where high recall is required, our scaling law defines what size of model should be trained at a given compute budget. In compute regimes less than two orders of magnitude below the final training run's size, we find that when holding the compute budget fixed it is desirable to use the "smallest" model trained on no more the final run's total token count as possible, and to frontload the data seen by this smaller model with sequences whose memorization would be undesirable in order to predict whether these sequences would be memorized by a final model.

### 5.4 Takeaways for Decision-Making

The experimental procedures we present in this paper are a step toward practical risk assessment for large-scale training runs. We hope that future work both makes such assessments much cheaper and accurate, and that more practitioners adopt such measures and choose to be up-front about which mitigations they apply when training large neural models.

## 6 Limitations and Future Work

Our work constitutes the first steps towards developing a way to predict what data will be memorized by a large language model before that model is trained, but has several limitations and opens opportunities for exciting future work. The most important of these are:

**Are we measuring the correct thing?** The definition of memorization we use is derived from what is currently popular in the academic literature, but it is unclear if it is the best definition to use. We believe $k$-extractibility to be well-grounded in privacy concerns of language models, but other metrics such as memorization score may be more natural when studying the *dynamics* of memorization in training.

**Does this generalize to other models?** We report our experiments on the Pythia suite, because it was the only current language modeling suite suitable for such work at the time of our research. However, this leaves open many questions about whether our results generalize to models trained with different hyperparameters or different data. We validate our experiments with replications on the deduplicated Pythia models and different hyperparameters in Appendix A and B, but no other model suite is suitable for replicating this analysis. This gap points to the need for more reproducible, public dataset model releases to advance research on memorization.

**What about the data contents?** Our work does not take the actual content of the training data into account at any point in time: we are looking exclusively at predicting memorization based on whether other cheaper models memorize the content. Future work looking into whether there are properties of the training text that predict memorization of that text could be quite illuminating.

## 7 Conclusion

We propose a novel setting for forecasting model memorization prior to train-time, while minimizing the compute required to make this forecast. We present analyses on the two most natural setups for extrapolation: using **fully-trained small models** and **partially-trained checkpoints of the final model** to compare and predict memorization of the final large model. We find that using much smaller models for this task is not viable, and that partial checkpoints of an existing model are similarly ineffective predictors of final memorization behavior when adjusted for cost. We derive a scaling law to find the optimal equi-compute predictor of non-memorization and are able to provide recommendations based on this law. We hope that our focus on prediction of the memorization of specific strings will be compelling for future study, and that our analyses inform deep learning practitioners on methods to understand and reduce memorization while training large language models.

# 8 Corrections

Due to an error in our analysis code, an earlier draft of this paper reported a substantially higher recall in Table 2. This draft of the paper features corrected numbers in that table and has adjusted the conclusions and discussion accordingly.

## Acknowledgments and Disclosure of Funding

This paper was made better by conversations with and feedback from many individuals not on the authorship list. Following EleutherAI's open science values [Phang et al., 2022], we shared early drafts of these results with the EleutherAI Interpretability Reading Group as well as the Discord server at large, garnering feedback from many people. We would like to acknowledge Nicholas Turner, Gurkenglas, and Amaru Cuba Gyllenste for identifying errors in our results and questioning our assumptions; Kyle O'Brien and Aviya Skowron for copy-editing; and Herbie Bradley, Nicholas Carlini, Katherine Lee, Naomi Saphra, and the EleutherAI Interpretability Reading Group for their thoughts, feedback, and advice.

We are grateful to Stability AI for providing the compute required to carry out our experiments.

Our work builds on top of the work of many teams at EleutherAI and within the broader open source community writ large. We'd especially like to recognize the GPT-NeoX [Andonian et al., 2021] team at EleutherAI whose library we used to measure memorization and the US AISI maintainers of the Hugging Face Hub whose infrastructure we used to host our data.

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

## A   Robustness to Thresholding Choices

In Section 3 and Section 4, we subset the data to the sequences with a "memorization score" of 1 (i.e., sequences that are fully memorized under previous works' definition). This approach labels all sequences with more than 32 tokens memorized as equally memorized, despite the fact that in reality some will have a much longer accurately reproduced continuation than others. In this section we explore whether that effects our results.

First, we examine the shape of the distribution of memorization scores. We had originally assumed that the answer would be an (approximately) exponential distribution, under the assumption that LLMs had a constant "memorization rate" for correctly predicting each subsequent token. Our assumption was that this "memorization rate" would be based on model

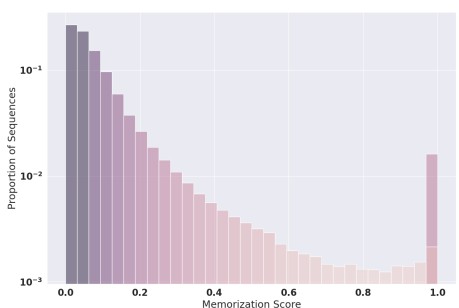

Figure 4: Distribution of memorization scores for 12B parameter model. For all upcoming sections in this paper, "memorized" is defined as $score = 1$.

size, and that it was the primary determinant of overall memorization score distribution. This would be potentially problematic for our study, as the 32-token memorized sequences would dominate the set of memorized sequences.

However, upon examining the distribution of memorization scores for the largest Pythia models, it was immediately clear that this cannot be the case. As shown in Figure 4, there is a very evident spike in the memorization score distribution at $score = 1$. Exponential distributions are thin-tailed distributions, and while they would have a spike at $score = 1$, it is not possible for them to have such a large spike. The effect shown in Figure 4 can only occur in *thick-tailed* distributions, such as the power law distribution.

This is a good sign for our analysis, as it means that the typical memorized datapoint in fact has a much larger number of memorized tokens than the 32 token threshold we were worried about. We also replicate Figure 1b with the doubled threshold and find roughly the same results. We also find the same results rerunning our scaling laws plots in Figure 6.

| Model Size | Precision | Recall |
|---|---|---|
| 70M | 0.949 | 0.140 |
| 160M | 0.941 | 0.222 |
| 410M | 0.931 | 0.334 |
| 1.0B | 0.922 | 0.451 |
| 1.4B | 0.918 | 0.497 |
| 2.8B | 0.900 | 0.611 |
| 6.9B | 0.872 | 0.775 |
| 12B | — | — |

Figure 5: Precision and Recall when using each model to predict which sequences would be memorized by the 12B parameter model. This table requires twice as many tokens to match to be considered memorized, but otherwise is a replication of Figure 1b.

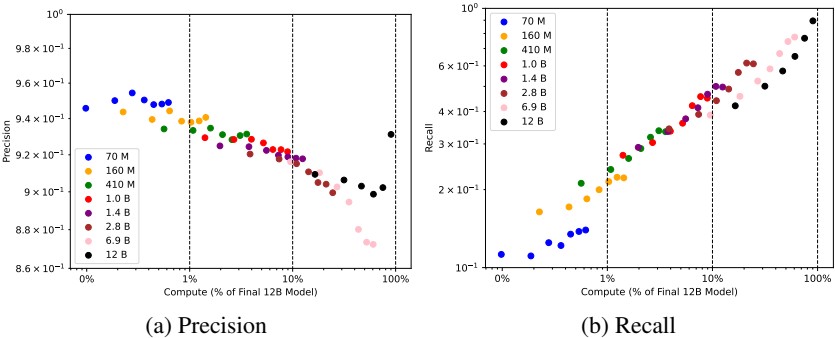

(a) Precision                    (b) Recall

Figure 6: Replication of Figure 3 with longer seqeunces.

# B   Robustness to Deduplication

In order to further confirm the validity of our analyses, we run our experiments on the Pythia (deduplicated) suite, which was trained on a deduplicated copy of the Pile [Gao et al., 2020] for 1.5 epochs. In keeping with the literature on deduplication and its connection with memorization [Lee et al., 2021, Kandpal et al., 2022], we observe that memorization is decreased for this set of models, albeit slightly (Figure 7). This may be due to the 1.5 epoch training setup we adopt offsetting the benefits of deduplicated data.

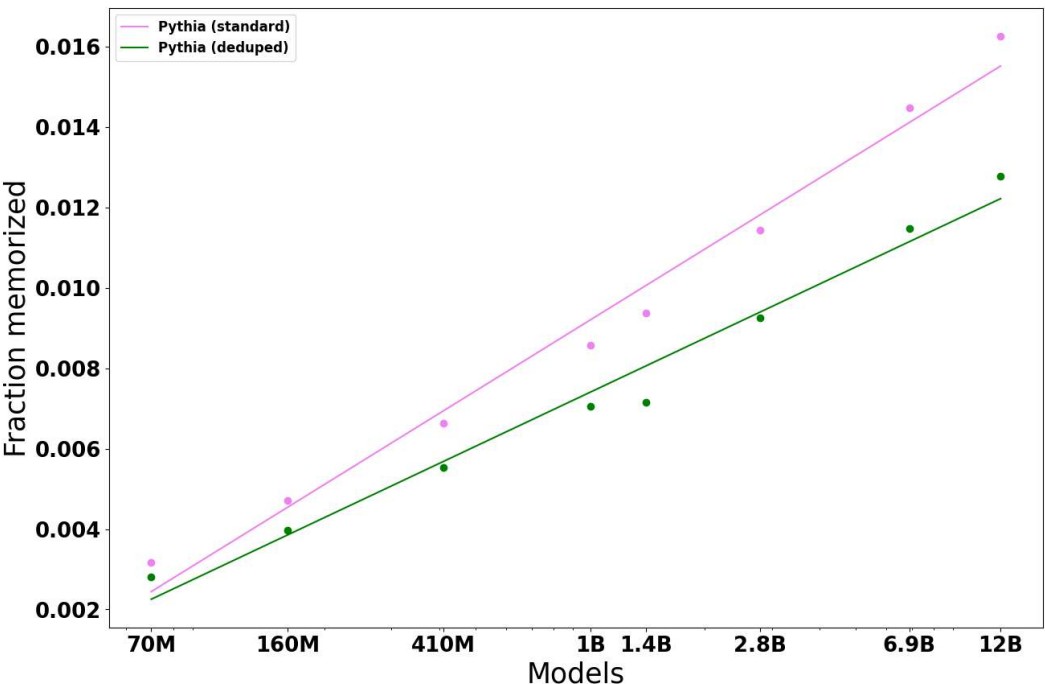

Figure 7: Fraction of all sequences memorized by both Pythia model suites. For example, Pythia-12B has memorized 1.62% of sequences. We can observe the deduplicated models memorize less of their dataset than their non-deduplicated counterparts.

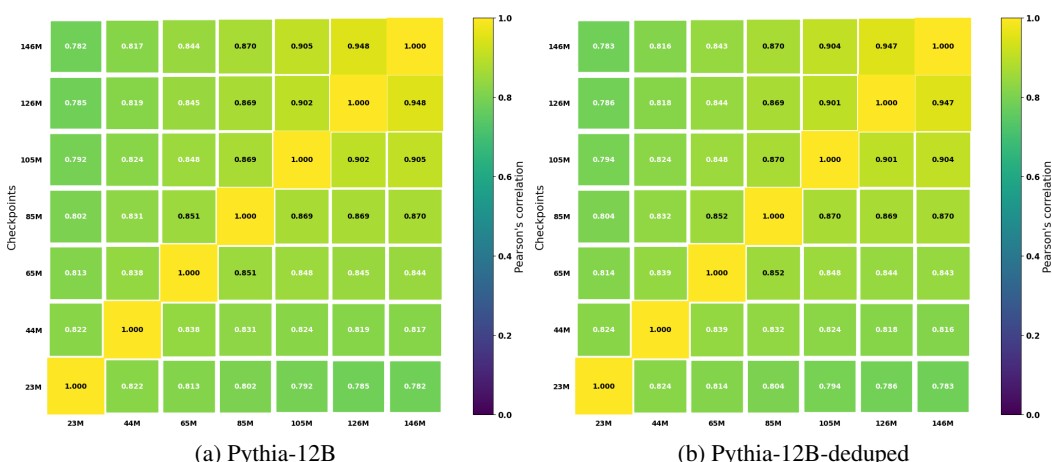

(a) Pythia-12B                                             (b) Pythia-12B-deduped

Figure 8: Inter-checkpoint correlations for memorization of Pythia-12B and Pythia-12B-deduped, respectively. Between the two sets of models, we observe extremely similar (though not fully identical) patterns in the correlations of their checkpoints.

| Model | Precision | Recall |
|---|---|---|
| Pythia-70M-deduped | 0.952 | 0.218 |
| Pythia-160M-deduped | 0.943 | 0.304 |
| Pythia-410M-deduped | 0.939 | 0.422 |
| Pythia-1.0B-deduped | 0.927 | 0.531 |
| Pythia-1.4B-deduped | 0.924 | 0.535 |
| Pythia-2.8B-deduped | 0.912 | 0.675 |
| Pythia-6.9B-deduped | 0.891 | 0.807 |
| Pythia-12B-deduped | — | — |

Figure 9: Precision and recall when using each model to predict which sequences would be memorized by the 12B parameter model. Replicates Figure 1b.

| Seq Num | Precision | Recall |
|---|---|---|
| $23 \cdot 10^6$ | 0.920 | 0.523 |
| $44 \cdot 10^6$ | 0.917 | 0.595 |
| $65 \cdot 10^6$ | 0.915 | 0.658 |
| $85 \cdot 10^6$ | 0.915 | 0.724 |
| $105 \cdot 10^6$ | 0.922 | 0.820 |
| $126 \cdot 10^6$ | 0.949 | 0.920 |
| $146 \cdot 10^6$ | — | — |

Table 3: Precision and recall for predicting which sequences would be memorized by the fully-trained model from a partially-trained checkpoint, for Pythia-12B-deduped. The trends observed here match Table 2.

We replicate our analyses on the deduplicated models and find the same trends hold for our experiments on the Pythia-deduplicated models as do for the regular Pythia suite. Heatmap correlation results show the same conclusions (Figure 8), and we replicate precision and recall results from Figure 1b and Table 2 but on deduplicated models in Figure 9 and Table 3. We therefore believe our results to be reasonably robust across hyperparameters and engineer train-time choices, but hope that future work may replicate some of our findings on entirely distinct corpora.

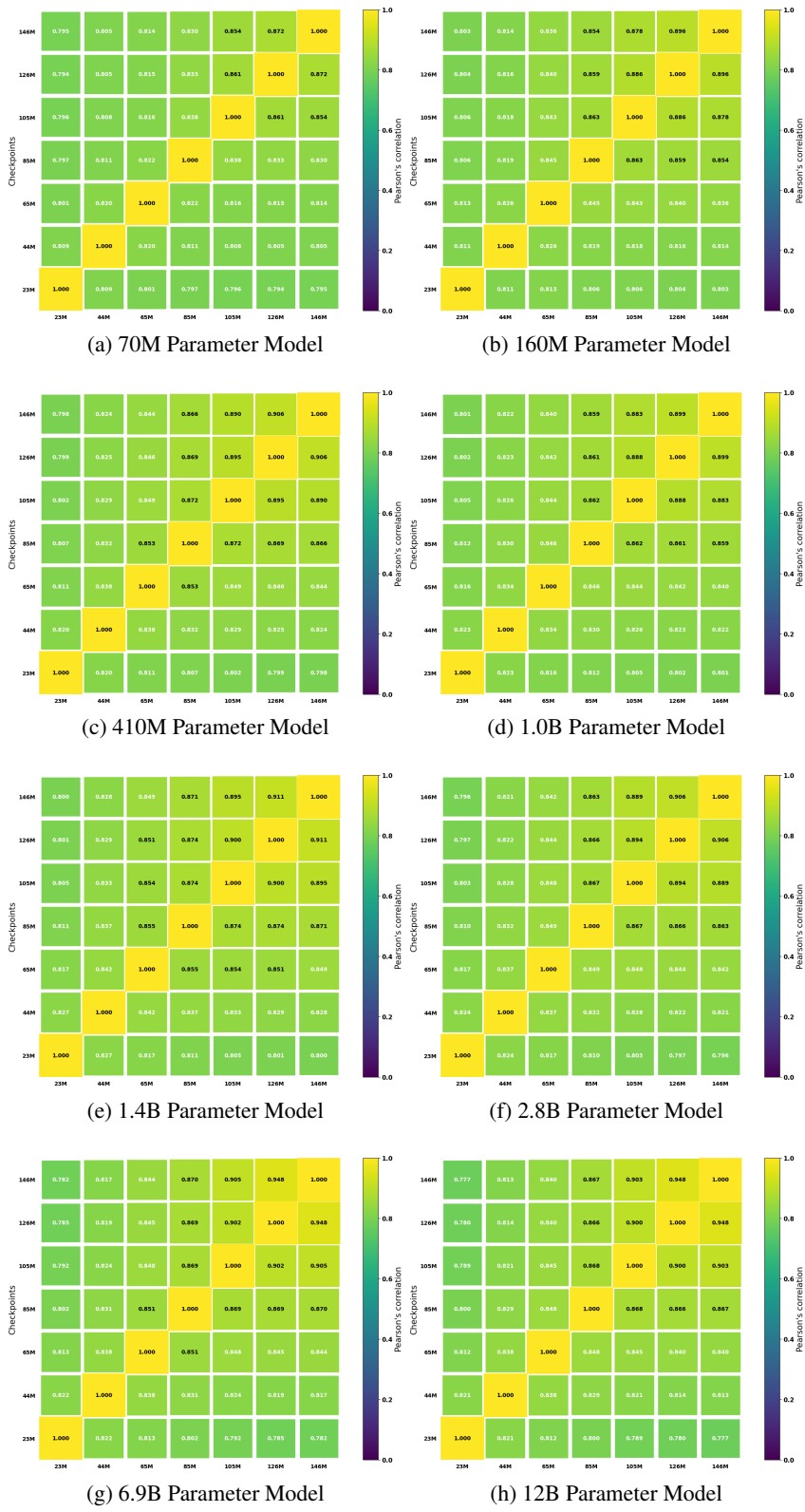

Figure 10: Heat maps visualizing the correlations between which sequences are memorized by different checkpoints.

## C  Author Contributions

**Stella Biderman**  Conceived, organized, and lead the project, and wrote the paper.

**USVSN Sai Prashanth**  Implemented and carried out the evaluation of memorization of pretraining strings.

**Lintang Sutawika**  Analyzed and interpreted the precision and recall results and plotted data.

**Hailey Schoelkopf**  Carried out the evaluation of memorization of pretraining strings, performed the robustness evaluation, found and fixed several bugs in our code, and wrote the paper.

**Quentin Anthony**  Analyzed and interpreted the results and wrote the paper.

**Shivanshu Purohit**  Optimized the implementation and assisted with carrying out the evaluation of memorization of pretraining strings.

**Edward Raff**  Designed the experiments, interpreted the results and wrote the paper.

