# OpenReview forum: "Emergent and Predictable Memorization in Large Language Models"
_NeurIPS.cc/2023/Conference — NeurIPS 2023 poster_

### Official Review · Reviewer_bdyA · 2023-07-03

**Soundness:** 3 good
**Presentation:** 4 excellent
**Contribution:** 3 good
**Rating:** 8
**Confidence:** 4

**Summary:**

This work presents an empirical study on predicting the memorization of examples in large language models. Specifically, using the Pythia model, they test two ideas for memorization prediction: (1) prediction via smaller models, and (2) prediction via partially trained models. Through extensive experiments on models with different scales, this work shows that while both method achieves non-trivial results, neither of them can reliably predict memorization with high recall. Therefore, this paper also includes experiments on selecting the best method w.r.t recall and discovers interesting trends.

**Strengths:**

1.	This work is an extensive empirical study on an important and timely topic.
2.	The analysis conducted in this paper are highly connected to real use cases and the findings also lead to practical suggestions.
3.	This is a very well-written paper with clear motivation, clean organization, and an honest, insightful limitations/future work section.
4.	I find the additional analysis in the appendix to be very interesting and can add significant value to this work. I would encourage the authors to move them to the main paper especially considering the current main paper is not very long.


**Weaknesses:**

The limitations section in the main paper is insightful and honest. I agree with everything written there and do not think any of the points is a severe flaw.

1.	As the authors acknowledged in the limitations, the use of k-extractible and the memorization score simplifies the problem. In practice, the attacker may not need to extract exactly the same information as the training data.
2.	Also mentioned in the limitations, the analysis in this work does not touch data properties or data contents, which are important future works.
3.	Also mentioned in the limitation, despite the large amount of experiments done in this work, it is still unsure how the findings will generalize to other models and data.
4.	Minor presentation issue: I find the takeaway in LINE 308-310 hard to understand.


**Questions:**

1.	Why is there a 6 in Equation 2?
2.	For Figure 1 and Figure 3, are the correlations computed using memorization scores or are they computed using 0/1 values? I assume it’s the former, but I don’t think it has been clarified in the paper.
3.	Are the precisions and recalls computed at the token level or at the example level?
4.	I’m wondering what is the upper bound of the prediction is. In Figure 1, the values at the diagonal are all 1, which makes sense as they are the same model. However, what will that value be if the initialization seed or the data order are different?


**Limitations:**

The authors have included a great limitations section in Sec. 6 in the main paper.

---

> ### Author Rebuttal · Authors · 2023-08-10
>
> We thank the reviewer for their very thorough and positive comments on our work, and are excited that they find our work may be of immediate empirical usefulness!
>
> > As the authors acknowledged in the limitations, the use of k-extractible and the memorization score simplifies the problem. In practice, the attacker may not need to extract exactly the same information as the training data.
>
> We agree that such an extension would be very valuable and insightful! For our work, we limit ourselves to the k-extractible setting (+ continuous memorization score) as the majority of memorization literature focuses on this setting, but a more thorough understanding of what sorts of memorization are harmful versus benign would provide excellent direction to future work. We are currently following up this work by investigating the taxonomy of what sorts of data points are memorized.
>
> We are grateful you believe the robustness analyses in the appendix add value to the work, and if we are able to present them within the length limit, we will move these to the main body of the text.
>
> The derivation of Equation 2 can be found in [Kaplan et al. 2020, Section 2.1] (https://arxiv.org/pdf/2001.08361.pdf). In brief, there are two operations per parameter per datum in a single forward pass and twice as many computations required to do the backward pass (as you operate on both the weights and the gradients)..
>
> Precision and recall are computed at the example level, and the correlation heatmaps are computed using the data subsetted to sequences that are fully memorized, though we see much the same results regardless of subsetting choices when computing these correlations. We will strive to improve the exposition and clarity of such captions and measurement details in the updated paper!
>
> >  In Figure 1, the values at the diagonal are all 1, which makes sense as they are the same model. However, what will that value be if the initialization seed or the data order are different?
>
> At the time of writing this paper, we did not have access to any models trained on different initialization or data seed that were otherwise the same. However, as Pythia-160m models trained on different data orderings + with different initializations were since released, we would be happy to rerun this analysis on such a model, though the time required to perform this experiment will not permit us to provide these results prior to the end of the review period.
>
> > Minor presentation issue: I find the takeaway in LINE 308-310 hard to understand.
>
> We will improve the presentation of this takeaway in an update to the paper! Our scaling law experiments calculate the memorization statistics *only on the first 1/7 of the corpus seen by every predictor we compare*, therefore assuming that if a practitioner knows of a subset of data they would like to predict memorization on, they should place that subset within the first 1/7 of the training data seen, so that predictions on it can be best measured. Else, measuring memorization on data appearing at the end of training would require a fully-trained predictor model to assess memorization under that predictor.
>
> We hope that these clarifications help increase your understanding of the paper, and intend to incorporate the feedback on presentation and exposition in an updated version of the paper if accepted. We appreciate your thorough feedback and believe it will lead the paper to become stronger as a result.

---

> > ### Comment · Reviewer_bdyA · 2023-08-18
> >
> > Thank the authors for the detailed clarification! I think this paper is an empirically solid analysis paper and can provide useful insights for future works. I will keep my score unchanged.

---

### Official Review · Reviewer_5bNj · 2023-07-06

**Soundness:** 2 fair
**Presentation:** 3 good
**Contribution:** 3 good
**Rating:** 6
**Confidence:** 3

**Summary:**

This paper discusses the issue of memorization in large language models, which can lead to the output of sensitive data. The authors propose to predict which sequences will be memorized by partially-trained checkpoints. They use the Pythia model in experiments and plot scaling laws for forecasting memorization, allowing them to provide equi-compute recommendations to maximize the reliability of such predictions.

**Strengths:**

1.The paper is well structured and addresses an important problem

2.The memorization score and extractibility are easy and effective


**Weaknesses:**

1.The authors set k=32 and evaluate the first 64 tokens from each sequence because of the computational costs,
which deviates greatly from the actual situation

2.They only use the Pythia models for evaluation

**Questions:**

In practical scenarios, how can we know in advance which data should be memorized well (such as factual knowledge) and which data should not be remembered (such as private personal data)? If we know in advance, the proposed method will no longer be needed; if we do not know, how can we improve the reliability of the model specifically after forecasting the distribution of the memorized data points?

**Limitations:**

The authors adequately discussed the limitations in the last section.

---

> ### Author Rebuttal · Authors · 2023-08-10
>
> Thank you for your review, and we are glad you find our posed memorization behavior forecasting problem important.
>
> > in practical scenarios, how can we know in advance which data should be memorized well (such as factual knowledge) and which data should not be remembered (such as private personal data)? If we know in advance, the proposed method will no longer be needed; if we do not know, how can we improve the reliability of the model specifically after forecasting the distribution of the memorized data points?
>
> We have received several requests for clarification on the interaction between our threat model and such filtering techniques. We apologize for the confusion our paper has caused, and we have responded to these questions in our general response.
>
> > 1.The authors set k=32 and evaluate the first 64 tokens from each sequence because of the computational costs, which deviates greatly from the actual situation
>
> We evaluate additionally on settings in which we require 64 tokens to be predicted accurately following a prompt of the first 32 tokens in a sequence (Supplementary Material, Appendix A), and find that all our results+conclusions in the paper are maintained. This gives us reassurance that our findings are not overly determined by our bounding the length of memorized generations. However, we respectfully disagree that this “deviates greatly” from the real world. Researchers in the real world are also resource constrained, and we believe that memorizing only 64 tokens will be problematic in some situations.
>
> We believe that the narrative and description of what experiments we reran can be improved, so we intend to clarify the experiments that we ran (namely, rerunning all our experiments to determine which of the first 96-token sequences (rather than 64-token sequences) in a context window are 32-extractible).
>
> > 2.They only use the Pythia models for evaluation
>
> Please see Q3 and Q4 in our general response for more detail on why we were forced to accept this limitation, and the steps we took to assess whether our results may transfer to very different datasets.
>
> We thank the reviewer for engaging with and improving our paper, and hope that our response will assure them as to the usefulness and robustness of our experiments as run in the paper.

---

> > ### Comment · Reviewer_5bNj · 2023-08-18
> >
> > Thanks for the response. I will raise my score.

---

### Official Review · Reviewer_S833 · 2023-07-06

**Soundness:** 3 good
**Presentation:** 3 good
**Contribution:** 3 good
**Rating:** 5
**Confidence:** 4

**Summary:**

This paper introduces the difﬁculties of both forecasting a future model’s memorization of speciﬁc training data and forecasting the memorization behavior of fully-trained LLMs from partially-trained checkpoints. Scaling laws proposed for forecasting memorization, and recommendations for maximizing forecast reliability given a set compute budget to make this prediction.

**Strengths:**

1、The core idea of  of  scaling laws for forecasting memorization makes sense to me, especially for larger models.

2、Proposed method is quite simple and works very efficient.

3、The paper is clearly motivated and written in a smooth and easy-to-read manner. For example, describes in detail the connections and differences with related work.


**Weaknesses:**

1、The essence of scaling laws is just the method that the author borrowed and tried instead of his own innovation.

2、The model used in the experiment in this paper is relatively uncommon and not convincing.

**Questions:**

Why not utilize LLaMA (Alpaca/Vicuna) or Falcon or other popular LLMs for experiments?

---

> ### Author Rebuttal · Authors · 2023-08-10
>
> Thank you for your feedback, and we are glad you feel the paper is well-written and effectively communicated the takeaways we intended.
>
> > The essence of scaling laws is just the method that the author borrowed and tried instead of his own innovation.
>
> We believe that our application of scaling laws to the memorization forecasting problem is a natural application and valuable to the community because:
> Scaling laws formalize the tradeoff between (compute) cost and performance on our novel problem setting. This is actively desirable, as it allows us to track progress on our defined forecasting task, where success is defined as strong predictive accuracy for a lower cost. They also allow practitioners to select freely along this cost-accuracy tradeoff.
> Following existing scaling law setups from the literature enables us to explore how a practitioner could *repurpose the small models they have already trained in a scaling law-fitting sweep* for this alternative purpose.
> This amortized cost is our target, as we hope that driving down the cost of memorization forecasts and countermeasures will encourage model trainers to place memorization and other security or privacy considerations as a first-class concern and a design decision on par in importance with model loss and downstream performance.
>
> Additionally, we hope that our application of scaling laws to this task will inspire other practitioners to consider how their typical scaling law studies can be designed such that *the compute cost of running the study/hyperparameter sweep* (the cost of making an extrapolated prediction) can be minimized while trading off a desired quality of extrapolation, as this angle is not often considered in the literature.
>
> > Why not utilize LLaMA (Alpaca/Vicuna) or Falcon or other popular LLMs for experiments?
>
> Please see section G3 of our general response for more detail on why we were unable to use other models for our experiments.
>
> > The model used in the experiment in this paper is relatively uncommon and not convincing.
>
> We would be happy to answer any questions about why we chose the prediction setups we did, and why we cast the memorization problem as binary classification in the paper.
> If this refers to our use of the Pythia models, we have responded to this in G3 and G4 of our general response, and hope that this may clear some things up.
>
> We are grateful that you found our paper “clearly motivated” and “written in a smooth and easy-to-read manner”. We hope that our work in this area will spur further study on forecasting and detecting memorization, and we hope our response will additionally convince the reviewer that the usage of “the essence of scaling laws” here is novel, and will help the community rally around this problem setting with a quantitative measure of success in future.

---

> > ### Comment · Reviewer_S833 · 2023-08-18
> >
> > Thank the authors for the detailed response! As the authors explained why they were unable to use other LLMs for the experiments, the method in this paper has great limitations in the era of LLM.  Perhaps the authors can clarify the value of the paper for  LLMs.

---

### Official Review · Reviewer_kL1W · 2023-07-08

**Soundness:** 3 good
**Presentation:** 3 good
**Contribution:** 3 good
**Rating:** 6
**Confidence:** 4

**Summary:**

The paper focuses on predicting memorization of specific data point prior to training the model. The main approach of the paper relies on scaling law, where the memorization property of a large model is predicted using a smaller model, or a partially trained model.
The findings are (1) small models are not good at predicting the memorization patterns of large LMs, causing very low recall score.
(2) using intermediate checkpoint to predict also attains low recall. (3) the prediction of memorization recall and precision does not follow the scaling law.

**Strengths:**

1. The setting of predicting whether a specific data is memorized is novel and interesting. The problem is well-motivated.
2. The approached tried in the paper are reasonable: predicting the memorization behavior of a large model using a smaller LM, using a intermediate checkpoint, and fit a scaling law curve.



**Weaknesses:**

1.  none of the proposed approaches produce the desirable recall that's needed for this approach to be viable. Even though there is a scaling law pattern, the best recall still seem to be not good enough.

2. If the developer know which data is sensitive, can they just remove the data from training distribution, and how much would that hurt performance? If they can just remove these data without harming performance, the basic motivation of this problem might be questioned.

**Questions:**

see above.

---

> ### Author Rebuttal · Authors · 2023-08-10
>
> We thank the reviewer for their feedback on our work! We are grateful that you find the target of our work well-motivated and novel, and agree that it is an important target for future work to improve beyond our performance on the task we have defined.
>
> > If the developer know which data is sensitive, can they just remove the data from training distribution, and how much would that hurt performance? If they can just remove these data without harming performance, the basic motivation of this problem might be questioned.
>
> We understand that there may be more clarification needed for how we envision our experiments’ practical application and connection to our described threat model. We believe our method can also prove helpful in the setting where all known PII cases are filtered, but not every occurrence will be successfully caught: for more information, please see our general response, sections G1 and G2 for clarification regarding this threat model and assumed problem setting.
>
> > none of the proposed approaches produce the desirable recall that's needed for this approach to be viable. Even though there is a scaling law pattern, the best recall still seem to be not good enough.
>
> We agree that the recall scores, even at high compute cost, are below a comfortable threshold for use by a practitioner or model trainer. We believe this is still a valuable result to report because
> It demonstrates that the problem of memorization forecasting is a difficult one, with much room for progress in absolute and equi-cost terms from the community. It appears to be quite a bit more difficult than the standard loss-prediction target of scaling laws, due to the intended prediction of fine-grained behavior.
> We believe that our simple approaches are the natural first steps, as they are such that practitioners might already produce lower-cost artifacts (smaller or partial model checkpoints) that can be applied via the same methods that are tested in our paper. It was not clear a priori that these natural approaches would be unsuccessful.
> Our work does not consider more advanced predictors, such as, for instance, training a small model for many epochs on a subset of the to-be-trained large model’s dataset. A limitation of using small models’ memorized sequences as predictors is that small models simply memorize less of their data than larger ones, which repeatedly training on the harmful-to-memorize subset might circumvent by intentionally overfitting to that data, memorizing similar quantities to a large model.
>
> We hope that this clears up some of the questions that you have had in reading our paper, and would be more than happy to discuss further with you.

---

### Official Review · Reviewer_c2gw · 2023-07-09

**Soundness:** 3 good
**Presentation:** 3 good
**Contribution:** 2 fair
**Rating:** 6
**Confidence:** 4

**Summary:**

This work empirically predict the memorization behaviors as measured by k-extractible and memorization scores. They tested on two settings 1) use small model to predict the memorization behavior or larger model, 2) use partially trained model to predict the fully trained model. The main finding is that 2) is has much higher correlation than 1). Also included scaling laws and discussions on emergent memorization.

**Strengths:**

* Clearly motivated and clean experiments with clear conclusions
* Well written and flows well
* interesting scaling law for emergent memorization

A clean paper with a clear message, addressing an important enough problem should be accepted.

**Weaknesses:**

* Not too obvious if the methods are good enough for practitioners to predict undesirable memorization due to lack of absolute metrics, in particular with respect to PII or something else people care about. Seems like random sequences were used in the experiments, which may have different behavior than PII
* Only tested on Pythia model suite and limited range of data (partly addressed in limitations).


**Questions:**

The heatmap is figure 3 is a wash, some variant of the Hinton diagram might show such correlation more clearly than color scales.

**Limitations:**

Yes

---

> ### Author Rebuttal · Authors · 2023-08-10
>
> Thank you for your review. We are grateful that you feel our problem setting is important and that the paper contains “clearly motivated and clean experiments with clear conclusions”.
>
> >  The heatmap is figure 3 is a wash, some variant of the Hinton diagram might show such correlation more clearly than color scales.
>
> We appreciate the feedback regarding how to communicate our results more clearly! We will regenerate the heatmaps as Hinton diagrams.
>
> > Not too obvious if the methods are good enough for practitioners to predict undesirable memorization due to lack of absolute metrics, in particular with respect to PII or something else people care about. Seems like random sequences were used in the experiments, which may have different behavior than PII
>
> You are correct that we studied randomly distributed sequences. This was a deliberate decision to limit the scope of our investigation because we were tackling a problem that has never been studied before. While PII is a motivating example, in this paper we are not studying memorization of PII specifically. We already have follow-up projects in the works studying how taking semantic information into account changes this analysis, but view this paper as an essential first step and an interesting self-contained result.
>
> > Only tested on Pythia model suite and limited range of data (partly addressed in limitations).
>
> We have discussed this further in G3 and G4 of our general response!
>
> We thank you for improving our visualization of our results and believe this will improve the usefulness of the paper to a reader, and your questions have been a helpful opportunity to assess which points are not adequately discussed in the paper body or limitation section. We hope that these further points will help answer your questions and clarify the motivation of the experiments we chose to perform in our work.

---

> > ### Comment · Reviewer_c2gw · 2023-08-18
> >
> > Read the response. The point on using only Pythia models are reasonable, but you can guess which data was included for most of the other models. Llama also says which data were included even if the data was not released. For the diagram, make the change only if you agree.
> >
> > Keeping the score.

---

### Author Rebuttal · Authors · 2023-08-10

We thank all reviewers for their valuable feedback on our work. We have worked to revise the presentation of the paper where recommended, and hope to provide this updated version in the final camera-ready submission. We would like to clarify some details regarding the application of our prediction method in practice where either PII can be filtered from the dataset prior to training, or a set of “known-harmful to memorize” sequences are not already known, and our threat model in those conditions:

G1: Why can’t we simply remove all harmful-to-memorize / PII-containing data from training datasets beforehand?

PII can be filtered from LLM training datasets via heuristic filters, or more advanced methods like [classifier-based approaches](https://arxiv.org/abs/2305.06161). However, these filtering methods are not perfect, and may fail to detect certain instances of PII that end up not filtered from the dataset.

Additionally, there are also non-PII data that are undesirable to memorize but desirable to train on. Copyrighted books form a substantial portion of the training corpus of a wide variety of models, including GPT-NeoX, Pythia, LLaMA, LLaMA 2, GPT-3, GPT-3, and many more. The current dominant thinking is that this is essential to achieving high quality performance (Longpre et al., 2023), but there are not nearly as many license-compliant books available as is standard for training models (Brown et al., 2020; Gao et al., 2020; Touvron et al., 2023). While the ethical and legal intricacies of permissible training data is outside the scope of this paper, we can confidently say descriptively that many organizations do in fact train on copyrighted text and [are currently being sued for it](https://www.theverge.com/2023/7/9/23788741/sarah-silverman-openai-meta-chatgpt-llama-copyright-infringement-chatbots-artificial-intelligence-ai). In particular, the lawsuit about [Copilot](https://www.theverge.com/2022/11/8/23446821/microsoft-openai-github-copilot-class-action-lawsuit-ai-copyright-violation-training-data) where the exact text of the model generations are explicitly the key issue.

Brown, Tom, et al. "Language models are few-shot learners." Advances in neural information processing systems 33 (2020): 1877-1901.
Gao, Leo, et al. "The pile: An 800gb dataset of diverse text for language modeling." arXiv preprint arXiv:2101.00027 (2020).
Longpre, Shayne, et al. "A Pretrainer's Guide to Training Data: Measuring the Effects of Data Age, Domain Coverage, Quality, & Toxicity." arXiv preprint arXiv:2305.13169 (2023).
Touvron, Hugo, et al. "Llama 2: Open foundation and fine-tuned chat models." arXiv preprint arXiv:2307.09288 (2023).

G2: How can our method (forecasting memorization with a small model) be applied in the absence of a known-bad-to-memorize data subset ahead of time?

One adaptation of our threat model (which takes into account the separation of intended and unintended memorization) is to perform a prediction of what data will be memorized, and *examine that set of predicted-to-be-memorized examples* for cases of PII by hand, as it is much smaller than the entire dataset. This extra PII-detection is a scenario in which high precision is valuable, so that any datapoints surfaced by the predictor are likely to be memorized by the final model and so this can serve as an early warning of undesirable memorization behavior.

We thus believe that even in the absence of a subset known to be harmful to memorize, our method can be applied as a “quality check” that confirms the absence of unintended memorization/overfitting/misgeneralization in a smaller preliminary model. Our strong results on precision do seem to indicate some quantity of data that is memorized across many different models. We would like to note that we are currently exploring the topic of the characteristics of the sequences memorized by our models, to assess the rate of occurrence of PII in memorization, what other properties are commonly held by memorized sequences, and other characteristics.

G3: Why did you not also use [Falcon/LLaMA/other non-Pythia model suites]?

We would be ecstatic to be able to run our analyses on LLaMA, Falcon, or other LLMs trained on other datasets. Unfortunately, the authors of those models do not release sufficient data to study questions relating to memorization of training data. Our analysis requires having access to the pretraining dataset, which we do not have for LLaMA, LLaMA 2, or Falcon. Additionally, some of our experiments require having the data *in the same order it was seen by the model*, or partially trained model checkpoints  The Pythia models are the only models known to us with a fully reproducible training data loader and possess far more partially trained checkpoints than any comparable model suite.

Finally, the Falcon models were impossible for us to compare to as they came out after the submission deadline for this conference.


G4: Do you expect these results to transfer to models trained on other datasets?

We hope so. In an effort to reduce doubts that our results would not transfer across different model training corpora, we replicated our experiments on the Pythia-deduplicated models trained on a deduplicated variant of the Pile and found the same conclusions. We hope that further work  either compare or allow us to subsequently compare results on models with highly differing training corpora, and note that the nonexistence of model suites outside of Pythia fitting our experimental requirements of:
Publicly available training data
Known, consistent data ordering
Varied model sizing
prevented us from evaluating any other non-Pythia models in this work.

We hope that these clarifications, along with other improvements to its presentation and organization, will increase the quality and usefulness to the community of our work, and would be happy to answer any further questions.

---

### Decision · Program_Chairs · 2023-09-21

**Decision:**

Accept (poster)

**Comment:**

The work empirically studies how to predict the memorization behavior of LLMs. It shows that using a partially trained model to predict memorization is more effective than using a small model.

Reviewers recognize the importance of the problem as well the empirical study and insights provided by the work.

Most of the concerns were addressed by the author in the rebuttal. The final scores are unanimous accept (5, 6, 6, 6, 8).

Please add the additional discussions, especially the ones regarding generalization of the method to other models and datasets in the final version.